# Transparent and attachable ionic communicators based on self-cleanable triboelectric nanogenerators

Younghoon Lee[1], Seung Hee Cha[1], Yong-Woo Kim[1], Dukhyun Choi[2] & Jeong-Yun Sun [1,3]

Human–machine interfaces have benefited from the advent of wireless sensor networks and the internet of things, but rely on wearable/attachable electronics exhibiting stretchability, biocompatibility, and transmittance. Limited by weight and volume, wearable devices should be energy efficient and even self-powered. Here, we report practical approaches for obtaining a stably self-cleanable, transparent and attachable ionic communicator based on triboelectric nanogenerators. The communicator can be easily applied on human skin due to softness and chemically anchored robust layers. It functions as a means of real-time communication between humans and machines. Surface functionalization on the communicator by (hepta-decafluoro-1,1,2,2-tetrahydrodecyl)trichlorosilane improves sensitivity and makes the communicator electrically and optically stable due to the self-cleaning effect without sacrificing transmittance. This research may benefit the potential development of attachable ionics, self-powered sensor networks, and monitoring systems for biomechanical motion.

[1] Department of Material Science and Engineering, Seoul National University, Seoul 151-742, South Korea. [2] Department of Mechanical Engineering, College of Engineering, Kyung Hee University, Seocheon-dong, Giheung-gu, Yongin-si 446-701, Korea. [3] Research Institute of Advanced Materials (RIAM), Seoul National University, Seoul 151-744, South Korea. Correspondence and requests for materials should be addressed to J.-Y.S. (email: jysun@snu.ac.kr)

With the advent of sensor networks at the interface between humans and machines[1], wireless and wearable/attachable electronics have emerged, including smart watch/glasses[2] or electronic/ionic skin that mimics the properties of human skin[3,4]. Several challenges are raised for devices that are desired to become smaller/lighter, flexible/ stretchable, and even transparent, while still requiring power sources. Batteries are the most commercialized portable power sources, but are relatively heavy and face critical lifetime limitations if they are made smaller or flexible[5]. As an alternative, harvesting mechanical energy from human motions is considered to be an attractive approach to meet the growing need for a wearable and lightweight power supplcy[6].

There have been extensive efforts to harvest mechanical energy based on piezoelectric, electromagnetic, and electrostatic effects[7]. Recently, triboelectric nanogenerators (TENGs), based on coupling the effects of contact electrification and electrostatic induction[8], have been achieving rapid progress as a sustainable power source with various advantages of simple structure (basically a couple of contact materials and electrodes), low weight, flexibility, and high energy conversion efficiency[9] for use in practical devices. Since the first report by Wang and colleagues in 2012[10], there has been an explosive development for harvesting various types of mechanical energy from activities such as walking[11], talking[12], and typing[13] for wearable electronics. In addition, surface treatment, which is a key for maximizing contact electrification and higher output power, has enabled TENGs to come close to commercialization[14]. Although there has been an increased interest in harvesting human motion without reducing degree of freedom (DOF) and transmittance, these applications have been struggling to provide both transmittance and stretchability due to the absence of proper electrodes. Using electrodes based on graphene[15], nanowire (NW)[16], and indium tin oxide (ITO)[17], there have been valuable efforts to increase both DOF and transmittance of TENGs for wearable electronics, but TENGs still suffer from a lack of stretchability and transmittance.

Hydrogel, which is the aggregation of crosslinked polymer networks in water[18], has been highlighted because of its promising properties such as biocompatibility[19], solubility[20], and so on. Specifically, a recent research by Sun and colleagues has greatly broadened the applications of hydrogel with extraordinary mechanical[21], optical, and electrical[22] properties, enabling its application to new functions, such as electrodes, actuators[22], and sensors[23]. These fascinating properties make hydrogels a promising material type for various research fields.

The use of a hydrogel could help solve the challenge of providing both high stretchability and transmittance in TENGs. Previous efforts have been made for the application of hydrogel to contact materials[24] or electrodes[25] of TENGs. Furthermore, Wang et al. recently demonstrated the possibility of obtaining stretchable and transparent TENGs[26]. However, the challenges of creating a stable adhesion of the hydrogel–elastomer layers, a low stickiness of contact surfaces for smooth working and anti-contamination, and a high output performance have not been met. These issues need to be addressed in order to enable practical application in our daily lives.

Here, we secure the robust adhesion by chemical bonding between the contact material polydimethylsiloxane (PDMS) and hydrogel, which is the electrode with benzophenone treatments. Additionally, considering that TENGs operate based on continuative contact or rubbing, sticky contact surfaces cause easy contamination. It is necessary to implement an anti-contaminated contact surface for maximizing transparency and electrical output performance. We explore optically, mechanically, and electrically stable TENGs that adopt self-cleanability with a high DOF. Without sacrificing the transmittance,

functionalization with (heptadecafluoro-1,1,2,2-tetrahydrodecyl) trichlorosilane (HDFS) ensures that the contact surface, which could otherwise become contaminated, is clean. The electrical output is increased and an antisticking effect[27] is implemented. In addition, in order to acquire both transparency and strong chemical bonding, we have investigated the relationship between the gelation time on the elastomer and the interface transmittance during chemically strong bonding. We further explore applications for not only harvesting finger touches for electricity but also for self-cleanable, transparent, and attachable ionic communicators (STAICs). Combination of touches of five fingers with thimble-type STAICs can express words to wireless electronics.

## Results

**STAICs with high mechanical reliability**. PDMS and hydrogel have extremely different affinities to water, resulting in weak physical bonding[28]. A mismatch of Young's modulus can cause defects between layers. For high reliability, it is essential to chemically anchor these to each other. We explored mechanically stable STAICs that are chemically bonded between the hydrogel and the contact material for high reliability. By applying benzophenone treatment shown in Fig. 1a, we ensured that the hydrogel is covalently and robustly anchored to the elastomer, as shown in the upper schematic of Fig. 1b (see Methods for details). Also, in order to maximize contact electrifications[14,29,30] resulting in high output power, many researchers have investigated the effects of physical surface treatment, such as nano/microstructure. However, this led to a decrease of transmittance[14] and durability[29] because of diffused reflection and wear, respectively. Thus, not only should the contact electrification be maximized, but also the transmittance. By using an HDFS chemical surface treatment, we formulated an approach to meet both of these requirements. The schematics for the fabrication procedure of STAICs are shown in Fig. 1a (see Methods for details of the fabrication steps). Perfluorination also helps to reduce reflectance of the surface[31]. Surface fluorination on STAICs by HDFS treatment, as shown in Fig. 1b, affects both the self-cleaning ability and stable electrical output power, which will be discussed later. Together with the non-treated STAICs surface used as a comparison, the functionalized STAICs were characterized by X-ray photoelectron spectroscopy (XPS) in order to elucidate the elemental compositions. As shown in Fig. 1c, a strong electron peak from the fluorine 1s orbital confirms the presence of fluorine on the STAICs, unlike for the non-treated STAICs. As shown in Fig. 1d–i, the STAICs are not only stretchable (Supplementary Movie 1), but can also be rolled, folded, twisted, and crumpled without any mechanical failure, which illustrates a high stretchability and softness. Furthermore, due to the chemical anchoring between benzophenone-treated PDMS and hydrogel, no delamination was observed during mechanical tests as shown in Fig. 1j. However, this chemical bonding exhibits a challenging side effect of becoming opaque at the chemically bonded interface between PDMS and hydrogel. One possible reason for this is that the high oxygen content included in the porous elastomer disturbs the gelation of hydrogel on PDMS. We provide the simple solution by controlling the gelation time to minimize the effect of oxygen. Fig. 1k and Supplementary Figure 1 distinctly show the difference in the transmittance as a function of the concentration of ammonium persulfate (APS) used as the thermal initiator and N, N,N',N'-tetramethylethylenediamine (TEMED) as the cross-linking accelerator, which easily controls the gelation time, resulting in a transmittance difference. An increase in APS concentration by 0.1wt%, 0.3wt%, 0.5wt%, and 0.8wt% led to an increase in transmittance at 550 nm by 30.2%, 55.8%, 95.2%, and 99.6%, respectively. It is simple but a vital point for ensuring

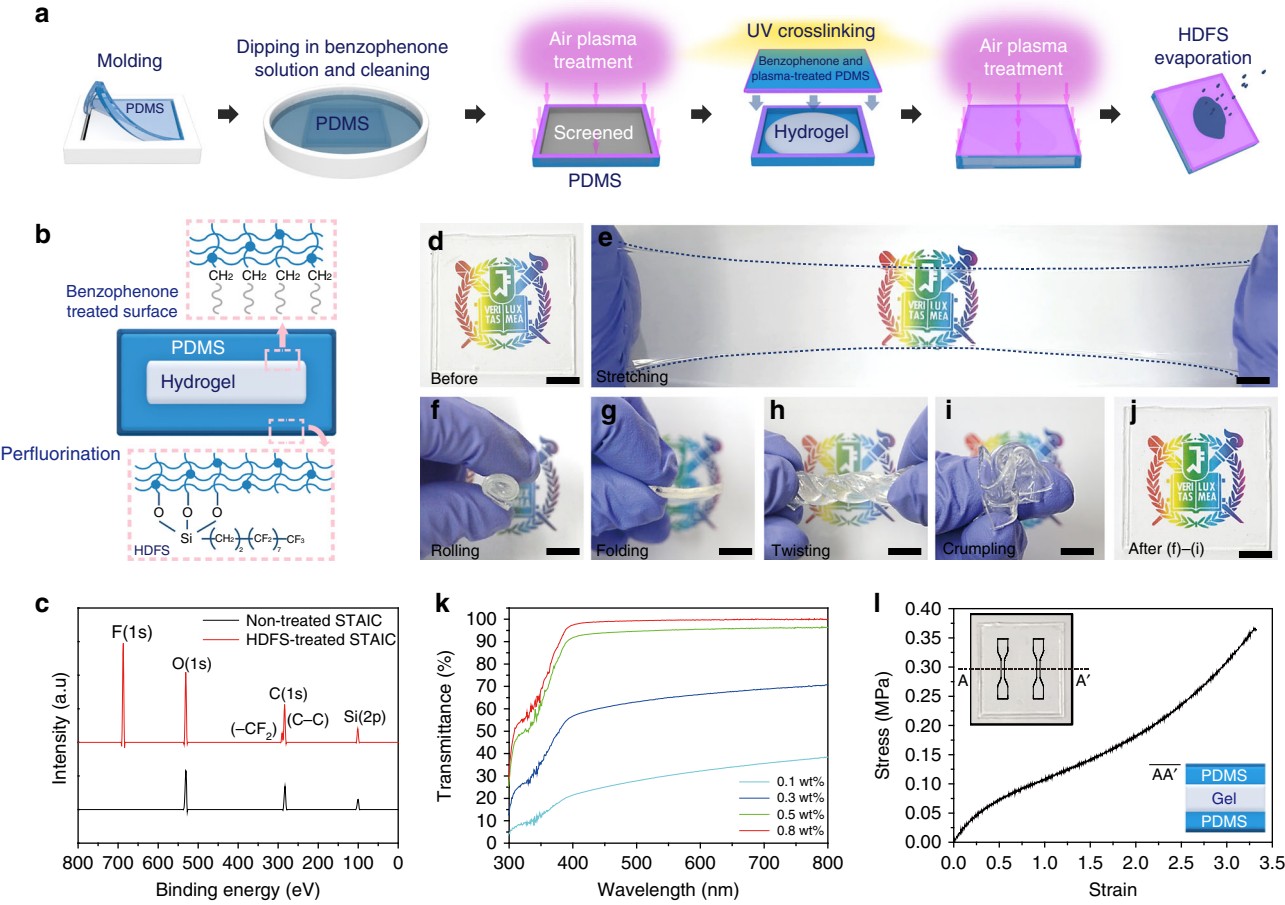

**Fig. 1** Transparent and deformable ionic communicators with high mechanical and optical reliability. **a** Fabrication process of self-cleanable, transparent and attachable ionic communicators (STAICs). **b** Cross-sectional structure of fabricated STAICs. Robust interfaces were formed between PDMS and hydrogel by applying chemical anchoring with benzophenone treatments (Heptacafluoro-1,1,2,2-tetrahydrodecyl)trichlorosilane (HDFS) coatings enhance triboelectric properties and provide self-cleaning ability. **c** X-ray photoelectron spectroscopy (XPS) analysis of HDFS-treated STAIC. **d, e** Fabricated STAIC was stretched with hands. **f-j** STAIC was subjected to different mechanical deformations sequentially and released. STAIC was **f** rolled, **g** folded, **h** twisted, **i** crumpled, and **j** released from the deformations. **k** Transmittance spectra according to various weight ratio of thermal initiator in STAIC. **l** Stress-strain curve of tensile test for a STAIC. (scale bar: 1 cm)

transmittance and a mechanically reliable, chemically anchored hydrogel–elastomer hybrid system. Although there are various types of PDMS, Sylgard-184 polymer has been widely used in the TENG field. However, Sylgard-184 has limited stretchability, up to 160%[32,33]. Therefore, we have used MS-1003 to enhance the stretchability up to 330% without sacrificing transmittance of STAICs. As-prepared STAICs were cut into dog-bone shapes (insets of Fig. 1l) by using a laser cutter for tensile testing. The typical nominal stress and strain curves are shown in Fig. 1l. Young's modulus of 238.8 kPa and a strain of 3.3 make STAICs attractive attachable devices.

**Triboelectric mechanism and characterization.** STAICs are operated based on contact electrification and electrostatic induction. Supplementary Figure 2 shows the details about the operational mechanism of STAICs in the case of a ground-assisted object for the single electrode mode. This mechanism shows that the hydrogel can be used in tribotronics as an ionic conductor, and that our system provides good contact materials for achieving reliable and high electrical output power. Fig. 2a shows the experimental process for electrical output performance. We chose an aluminum plate as a contact material. Different contact materials were investigated, as shown in Supplementary Fig. 3. Larger difference between electron affinities of contact materials led to higher triboelectric voltage outputs. In addition, the effect of time interval between consecutive contacts on the electrical output performance was investigated (Supplementary Fig. 4). Longer duration time after detachment caused lower output voltages, a result which comes from charge leakage. Fig. 2b, c show the electrical output performance depending on existence of HDFS treatment on STAICs. 24.5% of output voltage and 17.5% of output current were increased, which is possibly due to the fluorine layer that was formed by HDFS and had larger electron binding energy than PDMS[34]. Surface modification by HDFS also protects contact surfaces from contamination during multiple instances of contact, resulting in stable electrical outputs. Notably, STAICs showed very stable outputs over 25,600 cycles as shown in Fig. 2d. Supplementary Figure 5 shows that STAICs have good anti-dehydration ability under harsh conditions relative to the bare hydrogel due to the outer coating of PDMS. Bare hydrogel noticeably shrinks after 3 h in a desiccator and its weight decreases by 72.5%, while that of the STAIC decreases by only 2.8%. Electrical outputs of the STAICs were examined under various uniaxial strains as shown in Supplementary Fig. 6. When 200% of strain was applied, the output voltages slightly increased

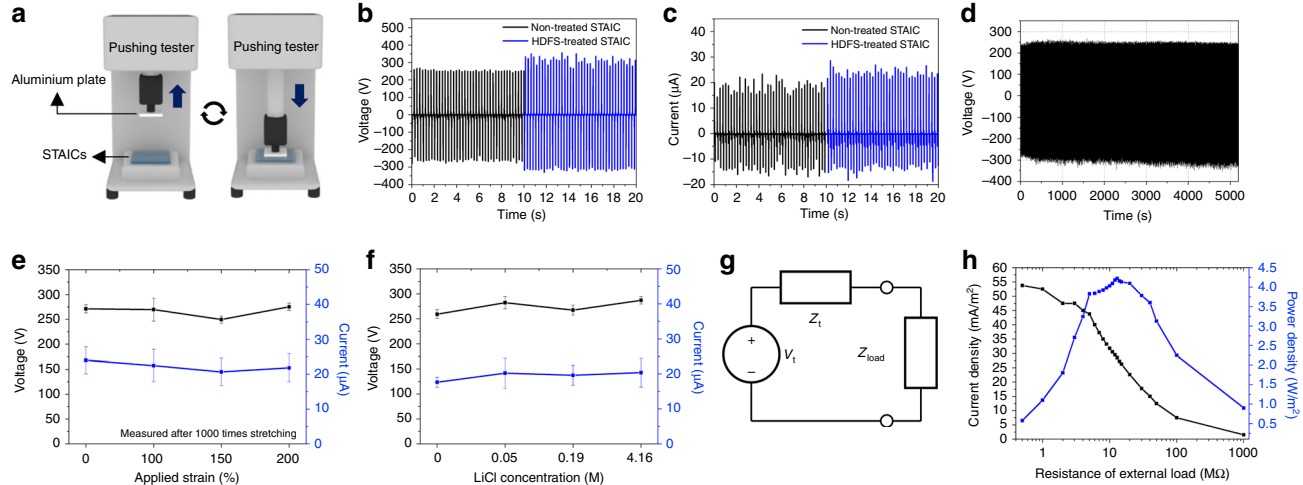

**Fig. 2** Triboelectric characterization. **a** Triboelectric performances were investigated with a pushing tester which can provide consistent contact area and speed (Heptadecafluoro-1,1,2,2-tetrahydrodecyl)trichlorosilane (HDFS) treatments for the self-cleanable, transparent and attachable ionic communicator (STAIC) enhanced **b** 24.5% of output voltage and **c** 17.5% of output current. **d** Durability of a STAIC was tested for 25,600 cycles. **e** Triboelectric outputs were measured after 1000 times of uniaxial stretching. 100%, 150% and 200% of strain were applied. **f** Effect of salt concentration on triboelectric outputs. **g** Thévenin equivalent circuit to deliver power to an external load ($Z_t$: Thévenin impedance, $V_t$: Thévenin voltage, $Z_{load}$: External load). **h** Output current density and output power density of a STAIC with various external loads. All error bars in the figure represent standard error of the mean of the data

from 252 to 277 V because of the reduced distance between the charged surface and the electrode. Degradations in electrical outputs by mechanical fatigues were tested after being strained 1000 times with strains of 100%, 150%, and 200%. The STAICs do not show any degradation in electrical output by mechanical fatigues as shown in Fig. 2e.

Resistance may be an issue for the use of hydrogel as an electrode of TENGs. Fig. 2f shows the effect of LiCl concentration on triboelectric performance. The results show that there was no significant electrical output changes caused by the change of LiCl concentration. The resistance of hydrogel which contains 0.01–1 M of LiCl has been reported in the previous paper[23], and the resistance was 360 Ω–22 kΩ (the size of a sample was 150 mm × 20 mm × 3 mm). Inherent impedance of STAICs can be calculated by Thévenin's theorem. Fig. 2g shows an equivalent Thévenin circuit of STAICs. To obtain a maximum power transfer point and estimate inherent impedance of STAICs, we varied external load. As shown in Fig. 2h, the output current decreased as the external resistance increased up to 1 GΩ. As a result, the maximum power density (4.22 W/m²) was obtained with an external load of 13 MΩ. The maximum power transfer happens when the external load is the same with inherent impedance of STAICs, so we can conclude that the inherent impedance of STAICs is 13 MΩ. Since this inherent impedance of the STAIC is much higher than the impedance of hydrogel, the performances of the STAICs were not influenced by the concentration of LiCl in the hydrogel. High transparency of STAICs allows the electrical power to be generated, without impeding the optical information. We have demonstrated the need for transparency by attaching a STAIC on top of a touch screen (Supplementary Fig. 7). A STAIC generated electrical power when the screen was touched, but did not hide the screen because the STAIC is highly transparent.

**STAICs with self-cleanability**. TENGs with polymer surfaces easily get contaminated during operation, resulting in a decrease of electrical output performances and transparency. Self-cleanability was achieved in STAICs by perfluorination through HDFS deposition with air plasma treatment. Fig. 3a, b show photographs of a static contact angle of a water droplet on non-

treated STAICs and on HDFS-treated STAICs. As shown in Fig. 3c, the static contact angle of water was not influenced by HDFS treatment with various plasma times. Fluorine functionalization normally makes the surface exhibit a higher contact angle[35], but because PDMS is adequately hydrophobic, the static contact angle was not significantly changed by the functionalization. Fig. 3d, e show advancing and receding contact angles on the non-treated (Fig. 3d) and HDFS-treated surfaces (Fig. 3e). The HDFS-treated surface showed higher receding contact angles making contact angle hysteresis smaller (Fig. 3f). Because a water droplet can be detached easily from a hydrophobic surface that has a high receding contact angle, HDFS treatments making the surface self-cleanable. The details about advancing and receding contact angles are plotted in Supplementary Fig. 8a, b. Fig. 3g shows the change of transmittance at the wavelength of 550 nm of the STAICs by HDFS treatment with various plasma times. The transmittance of STAICs was not influenced by the HDFS treatments and showed nearly 99% at all times. Compared to previous studies on modified surfaces with nano–micro patterning[14], this HDFS surface treatment has an advantage of retaining its transparency. The self-cleanability of STAICs was demonstrated by applying activated charcoal powder, which adheres well, as the simulated contaminant on each STAICs. Fig. 3h shows the cleaning behaviors of non-treated and the HDFS-treated STAICs according to the following procedure. In the initial state, photographs indicate that both the non-treated and treated STAICs show good transparency. By using ImageJ, photographs with dimensions of 30 mm by 30 mm were converted to 16-bit grayscale images to accurately show the contaminated (black) and uncontaminated (white) areas. An abundant amount of activated charcoal powder was spread on each horizontally placed STAIC. As shown in Fig. 3i, each STAIC was covered with activated charcoal powder and tilted by approximately 45° relative to the horizontal line. Approximately 5 mL of water in a syringe was shot twice onto each surface (Fig. 3j). Water droplets contacted the powder and removed contaminants by the rolling down. Subsequent water drops cleaned the surface, showing self-cleaning behavior that is generated by the HDFS treatment (Supplementary Movie 2). Followed by the water jet, each STAIC was cleaned with ethanol by ultrasonication for 10 s and 5 min.

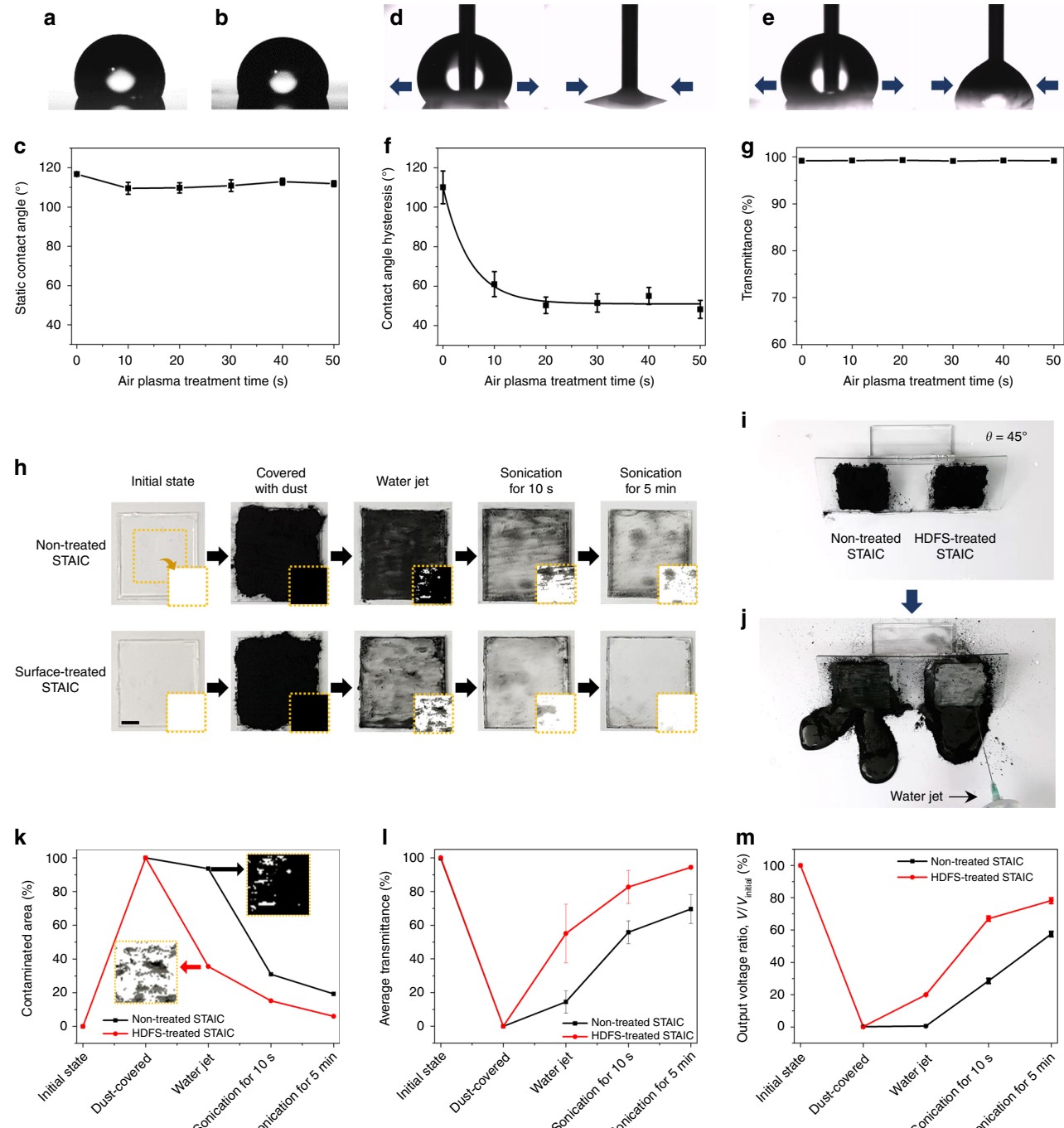

**Fig. 3** Self-cleanability of transparent and attachable ionic communicators. Static contact angles were observed for **a** non-treated surfaces and **b** (heptadecafluoro-1,1,2,2-tetrahydrodecyl)trichlorosilane (HDFS) treated surfaces. **c** Static contact angles were investigated for various plasma time. Advancing and receding contact angles measured with deionized water on **d** non-treated surfaces and **e** HDFS-treated surfaces. **f** Contact angle hysteresis was greatly influenced by HDFS treatments making the surface self-cleanable. **g** Transmittance of self-cleanable, transparent, and attachable ionic communicators (STAICs) for a green visible light (550 nm). **h** Self-cleanability of HDFS-treated STAICs was explored by cleaning surfaces which were contaminated by activated charcoal powders. **i, j** Demonstration of self-cleanability of STAICs. Dust on HDFS-treated STAIC was removed easily by a water jet. **k** Contaminated areas and **l** average transmittances at 550 nm were measured according to the processes. Ten points on STAICs were randomly explored to calculate the average transmittance. **m** Output voltages of STAICs were investigated according to the cleaning processes. All error bars in the figure represent standard error of the mean of the data (scale bar: 1 cm)

To quantitatively investigate the self-cleaning effect, the contaminated area on each surface was estimated by using ImageJ (Fig. 3k). After the water jet process for HDFS-treated STAICs, the contaminated area was reduced to as low as 35.6%p, compared to 93.6%p remaining in the case of non-treated STAICs.

After ultrasonication in ethanol for 10 s and 5 min, contamination areas of HDFS-treated STAICs were reduced to as low as 15.2%p and 6.0%p, compared to 31.0%p and 19.3%p in the case of non-treated STAICs. To further investigate the self-cleaning effect, UV−vis spectroscopy was used to study the transmittance

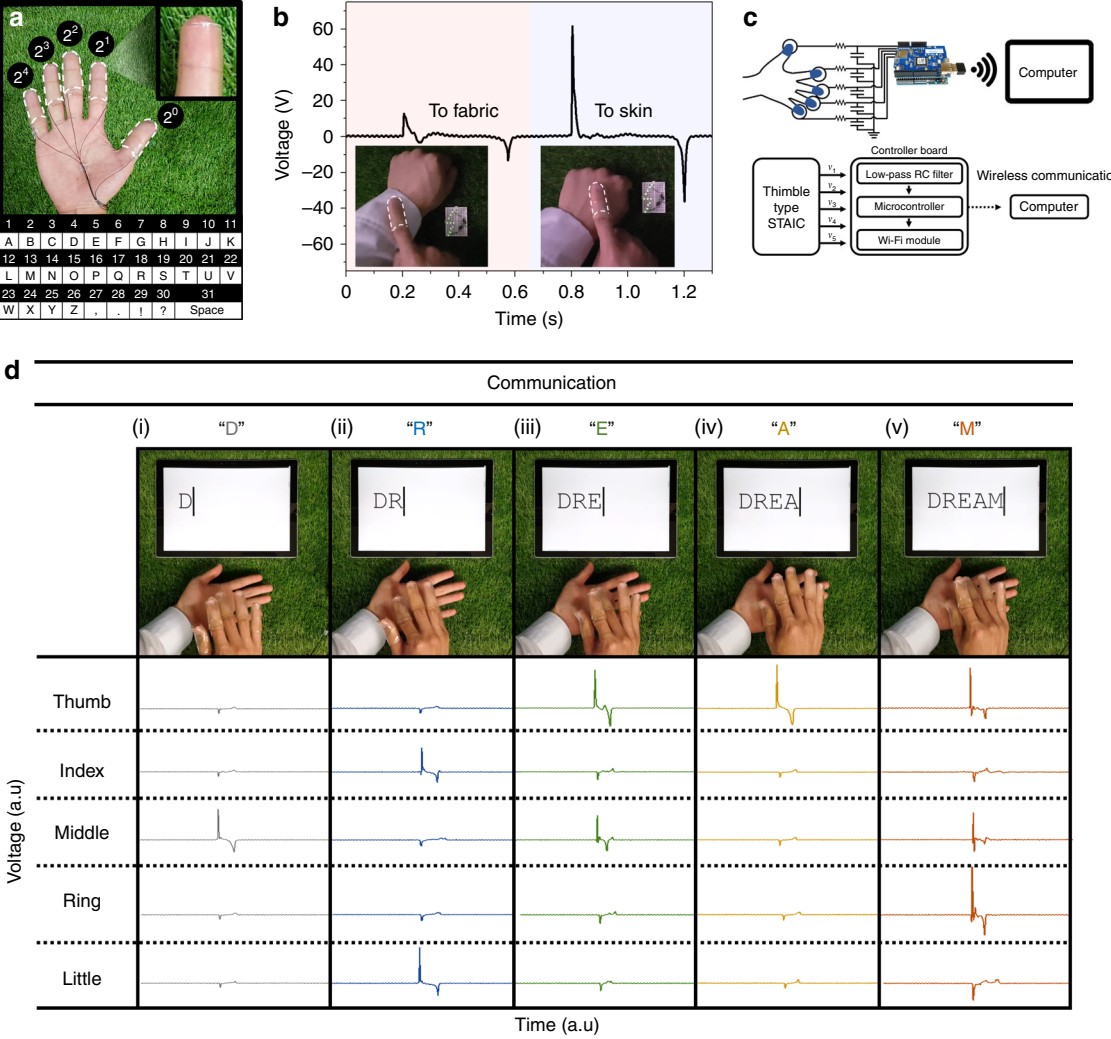

**Fig. 4** Wireless real-time communication. **a** Self-cleanable, transparent, and attachable ionic communicators (STAICs) were attached on the fingers and connected with wires to a controller board. A different order of a binary system was assigned to each finger, and the letters were pre-coded in the microcontroller. **b** Generated voltages of a STAIC by a gentle touch to a fabric and the skin. **c** Block diagram of the communicator based on STAICs. STAICs were connected to a controller board which contains an RC low-pass filter, a microcontroller and a Wi-Fi module. **d** Demonstration of real-time communication with STAICs. "DREAM" was typed using combinations of finger touches

of the STAICs (Fig. 3l). Average transmittance was obtained by randomly scanning selected points 10 times. After the water jet process for the HDFS-treated STAICs, average transmittance was increased to 55.1%p, compared to the increase of 14.5%p in the case of non-treated STAICs. After ultrasonication in ethanol for 10 s and 5 min, the average transmittance of HDFS-treated STAICs was increased to as high as 82.7%p and 94.4%p, compared to the increase of 55.8%p and 69.6%p in the case of non-treated STAICs. Moreover, since contaminated contact surface reduces triboelectric performance[36,37], output voltages of STAICs were investigated after each cleaning process, as shown in Fig. 3m. Due to the self-cleaning ability of HDFS-treated STAICs, the output voltage of HDFS-treated STAICs was enhanced to 19.4%p after the water jet, compared to remaining at 0.4%p for the non-treated STAICs. The electrical output performance was suitably recovered to as high as 67.0%p and 78.2%p after the further ultrasonication cleaning process, compared to the increase of 28.6%p and 57.5%p of non-treated STAICs. Thus, HDFS treatments on STAICs can provide optical and electrical stability.

**Wireless real-time communication based on STAICs.** A thimble-type wireless real-time communicator was produced for human–machine interfaces based on STAICs. STAICs were attached to fingers as shown in Fig. 4a. Thumb, index, middle, ring, and little fingers were coded to signal $2^0$, $2^1$, $2^2$, $2^3$, $2^4$ with a microcontroller for real-time communication, respectively. The combinations of the signals from each finger were interpreted to an alphabet as shown in the inset of Fig. 4a. Prior to the communication, we checked the triboelectric property of STAICs on a finger by touching skin and clothes. The STAICs successfully harvested electrical energy from a gentle touch to skin (61.6 V) and fabric (13.0 V) (Fig. 4b and Supplementary Movie 3). In addition, the effect of humidity on triboelectric performance was investigated in Supplementary Fig. 9. The output voltage decreased from about 270 to 50 V as the relative humidity increased from 20% to 80%. However, a STAIC can operate under a high relative humidity of 80%, because it can still generate detectable voltages. A circuit was then fabricated to make real-time communication between the STAICs and a wireless computer, as shown in Fig. 4c and Supplementary Fig. 10. STAICs were connected to a microcontroller through a low-pass RC filter to decrease electrical noise. The working flowchart of the real-time communicator is presented in Supplementary Fig. 11.

Fig. 4d shows a demonstration of a wireless real-time communicator based on STAICs. To input letter "D", a value of 4 is needed, which is given by a single touch of the middle finger ($2^2$) (Fig. 4d i). For writing "R", a value of 18 is needed, requiring simultaneous touch of the index ($2^1$) and little finger ($2^4$) (Fig. 4d ii). Writing "E", "A", and "M" is based on the same mechanism. The word "DREAM" was successfully typed with STAICs as shown in Supplementary Movie 4.

## Discussion

Highly transparent (99.6%) and stretchable (330%) STAICs were fabricated with mechanical reliability by chemical bonding between the ionic conductor and contact material, which is critical for the functionality and durability of hydrogel electronics. By controlling gelation time, both high transmittance and a robust hydrogel–PDMS interface were secured, preventing the light scattering on the interface of the chemically bonded layers. Additionally, HDFS surface treatment ensured an increased electrical output power (increase of output voltage by 24.5% and of output current by 17.5%) without sacrificing the transmittance in comparison with non-treated STAICs. Self-cleanability was attained by HDFS surface treatment, leading to recovered transmittance (40.6%p increase) and electrical output voltage (19.4%p increase) compared to the non-treated STAICs at the water jet step after simulated dust covering. Due to the stable transmittance and stretchability, STAICs served as a thimble-type, wireless, real-time, human–machine communicator by harvesting gentle touches with fingers. Remarkably, the attractive features of STAICs open up exciting opportunities not only for self-powered, transparent, attachable HMIs for wireless sensor networks but also for broader applications of stretchable ionics, soft robotics, and self-powered monitoring systems for biomechanical motion.

## Methods

**Materials and specimen preparation**. Unless otherwise indicated, the STAICs were made using lithium chloride-containing polyacrylamide hydrogel as the primary material. PDMS (Dow corning, MS-1003) was used as the contact material. Acrylamide (AAm; Sigma, A8887) and lithium chloride (LiCl; sigma, L4408) were used as base materials for the hydrogel and the ionic charge carrier. *N,N*-methylenebisacrylamide (MBAA; Sigma, M7279) was used as the cross-linking agent for the AAm gel. APS (Sigma, A9164) and *N,N,N′,N′*-tetra-methylethylenediamine (TEMED; Sigma, T7024) were used as the photoinitiator and accelerator for the gelation reactions, respectively. (Heptadecafluoro-1,1,2,2-tetrahydrodecyl)trichlorosilane (HDFS; JSI Silicone Co., H5060.1) was used as the surface fluorination agent. Activated charcoal (Sigma, 242276) was used as artificial dust. VHB 4910 (3 M) was used as the double-sided tape. A Pt electrode was used for the measurement of STAICs electrical characteristic and Ag coated Cu wire was used for the electrical circuit. Dog-bone-shaped specimens were cut with a laser cutting system (ULS, VLS 3.50) with the dimensions suggested by the ISO 37-4 standard for measuring stretchability.

**Fabrication of the hydrogel solution**. Unless otherwise indicated, the ionic hydrogel was synthesized by dissolving AAm monomer and LiCl in DI water. The molar concentrations of AAm and LiCl aqueous solutions were 4 and 16 M for all experiments. 0.05 wt% of the cross-linker MBAA, 0.7 wt% of the initiator APS and 0.35 wt% of the accelerator TEMED, with respect to the weight of the AAm monomer, were mixed and degassed.

**Fabrication of the STAICs**. PDMS (Dow corning, MS-1003) was prepared by molding the mixture of base and curing agents (5:1 by weight) followed with a 50 °C treatment in an oven for 24 h. Bottom PDMS was shaped with the wall near the rim with the dimensions of 0.5 mm and 1.0 mm (height × thickness) on 0.5 mm × 40 mm × 40 mm (height × length × width) and top PDMS was shaped with dimensions of 0.1 mm × 40 mm × 40 mm (width × length × thickness). We used acrylic as templates. The cured PDMS was dipped in 15 wt% benzophenone in ethanol solution for 10 min and cleaned with methanol to activate the elastomer surfaces[38]. To bond between the edges of top and bottom PDMS, the edges of bottom and top PDMSs were treated with air plasma for 10 s. The area that did not need to be treated was masked with a polyimide tape. Hydrogel solution was poured on the activated bottom PDMS and covered with top PDMS. Hydrogel forms chemical bond to the PDMSs by directly curing hydrogel precursor onto the benzophenone absorbed PDMS surfaces followed by 365 nm UV irradiation for an hour. The top STAIC surface was treated by air plasma for 30 s. Reactive oxygen radicals produced by air plasma attack the methyl groups ($Si–CH_3$) and substitutes them with silanol groups ($Si–OH$). A small amount of (heptadecafluoro-1,1,2,2-tetrahydrodecyl)trichlorosilane (JSI Silicone Co. Korea) dissolved in hexane (1:3) and OH- terminated STAICs were placed into the glass container, and then heated at 40 °C in an oven for 90 min. A self-assembled monolayer (SAM) was formed via the vapor. A Pt electrode was inserted through capsulated PDMS to the hydrogel to measure electrical characteristic of STAICs.

**Wireless real-time communication**. Prepared STAICs on transparent double-sided tape (Very High Bond; 3 M, 4910) were attached on each finger. Each wire from the STAICs was connected to the low-pass RC filter; 10 M ohm resistor and 1 nF capacitor were used. Through the low-pass RC filter, it was connected to a channel of the microcontroller (Arduino uno, Arduino Co) with a Wi-Fi shield (SOLLAE SYSTEMS Co, PHPoC). The microcontroller and the Wi-Fi shield were connected to a battery. The microcontroller was wirelessly connected to a computer (Microsoft, Surface Pro4).

**Characterization and measurement**. The electrical voltage and current of STAICs were measured by using an oscilloscope (1 MΩ, MDO3052, Tektronix) with a voltage probe (40 MΩ, TPP0850, Tektronix) and a low-noise current preamplifier (Stanford research systems, SR570), respectively. A pushing tester (Junil Tech, JIPT-100) was used to create vertical compressive force with an aluminum plate as a contact material. All tests were performed under the pressure of 37.5 kPa, a gap of 13 mm, and a frequency of 4 Hz. All measurements were performed after about 1000 pre-contacts to obtain a stable electric signal and were performed at room temperature. Impedance was measured using a Precision LCR Meter (Agilent, E4980A) (under input 1 V and 1 MHz). For the tensile test, a universal-testing-machine (Instron, 3343) was used, with the strain rate fixed at 60 mm/min. Cyclic stretching was performed during 1000 cycles (Z-tec, ZBT-300). The optical transmittance, of which reflectance was not considered, in the 300–800 nm range was measured by UV–vis spectroscopy (Agilent, Cary-60). Unless otherwise indicated, surface activation to form hydroxyl terminated of PDMS surfaces was performed with a plasma processing system (FEMTO SCIENCE, COVANCE-1MPR) under 200 W, 30 s, 15 sccm after rinsing with ethanol followed by $N_2$ blow drying. The contact angles between 10 μL drops of high-purity water and STAICs were measured using a contact angle analyzer (FEMTOFAB, Smart Drop) at room temperature. The average values of three measurements at different positions of the sample were used. For qualitative estimation of SAMs on STAICs, XPS data were monitored (KRATOS, AXIS-HSi). For analysis of the contamination area of the STAICs, an image processing program (ImageJ) was used.

**Data availability**. The data that support the findings of this study are available from the corresponding authors upon reasonable request.

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

## Acknowledgements

This work was supported by the National Research Foundation of Korea (NRF) Grant funded by the Korean Government (MSIP) (No. 2015R1A5A1037668). J.-Y.S. acknowledges the support from the Creative-Pioneering Researchers Program through Seoul National University (SNU). The authors thank Donghyun Seo for advice about surface wetting characteristics.

## Author contributions

Y.L. conceived the idea, analyzed the data, and wrote the main manuscript text. S.H.C. and Y.L. conducted the coding for the communicator. Y.-W.K. characterized the stretchability of the samples in the tensile test. D.C. provided advice for the research. J.-Y. S. supervised and conceived this study, and provided intellectual and technical guidance. All authors discussed the results and commented on the manuscript.

## Additional information

**Competing interests:** The authors declare no competing interests.

