## [Peer Review File · Nature Communications]

Reviewers' comments:

Reviewer #1 (Remarks to the Author):

See attachment.

Reviewer #2 (Remarks to the Author):

The author reported in the manuscript triboelectric nanogenerators. The device is transparent and can be easily cleaned. The property and performance have been investigated in detail. A self-powered communicator has also been fabricated and demonstrated. This work is important and of interests to others in the community. This reviewer supports the publication of this work, although a few questions are also listed below.

1. The author made a significant effort in studying the transparency of the device. It will be very important to explain why the transparency is important for the reported nanogenerator.
2. The nanogenerator worked in a single electrode mode. The contact material would be the other electrode. It will be helpful to study or discuss the effect of contact materials.
3. Again, the performance should be affected by the contact material (fabric, skin, or others). These materials are different in keep static charges. How does the energy conversion process change when time is considered. In other words, if the supplementary figure 2c stay for a longer time, the charges on the finger might changes, such that the charge transport (energy conversion) in supplementary figure 2d might be different.
4. The explanation of the self-powered communicator is not clear. What is the circuit? How the communication realized?

Reviewer #3 (Remarks to the Author):

See attachment.

Reviewer #1 (Remarks to the Author):

This manuscript reports a transparent and stretchable triboelectric nanogenerator as self-powered and attachable ionic communicators. The authors need to address the following concerns when submitting the revised version.

1. From the supplementary figure 2, a wire is connected to the hydrogel inside of the PDMS, but there is no wire connection shown in figure 1 (a). Can the authors explain how exactly did the wire connected? Is it simply inserted into STAIC after process flow in figure 1 (a)?
2. In figure 2 (e), triboelectric outputs were measured after 1000 times of uniaxial stretching, which demonstrates the stability of the STAICs after different strain conditions. However, have the authors considered the triboelectric output under different strain? Since it is attached to human skin, this kind of situation should also be taken into consideration.
3. For voltage measurement, the authors use an oscilloscope (Tektronix, MDO3052). What is the impedance of the measurement setup, since the impedance of TENG is high.
4. Supplementary Figure 3 shows that STAICs have good anti-dehydration ability in harsh conditions relative to the bare hydrogel, however we can still observe a 2% reduction of the weight ratio after 5 hours. Does that mean that PDMS is not enough to perfectly seal the hydrogel to prevent the water evaporation? If so, did the authors tested the lifetime of the STAICs under harsh environment?
5. It is better to label the static contact angle and receding contact angle in Fig. 3, for some readers may be not familiar with it.
6. The demonstration of wireless real-time communication is interesting, but the need for transparency in this application is not strong enough.
7. The supplementary Fig 2 does not give any novel information. It is well know phenomenon.
8. Please go through the manuscript more carefully since there are some grammar errors and typos, e.g. Page 10, Line 191: “this HDFS surface treatment has an advantage in its the transparency”; Page 10, Line 197: “By using ImageJ, photographs with dimensions of 30 by 30 mm” better be “30 mm by 30 mm”; sometimes “Fig. 3h iii” with “.” is used, sometimes “Fig 3h iv” is used, please be consistent; etc.
9. Figure 4 (c) shows the block diagram of self-powered communicator, in which the STAIC is connected to a controller board. Is the controller board powered by external source? If so, it might not be accurate to claim the whole system is self-powered. Please clearly illustrate this part in the manuscript. Besides, please provide the detail circuit diagram of the controller board in supplementary.
10. Sweat is always a critical issue the affects the performance of the human-machine interfaces. In that case, did the author investigate the effects of humidity on the triboelectric output? This phenomenon may lead to a high unstability of the device performance.

Reviewer #3 (Remarks to the Author):

This manuscript presents a novel transparent attachable ionic communicator driven by TENGs. The originality of this paper is significant, application of self-cleaning effect is smart. The result is interesting and promising for five fingers communicator.

However, some results are not well explained in the text. The following aspects should be revised before considering this manuscript for publication.

1. In the manuscript, the authors illustrate that hydrogel by HDFs treatment to provide high electrification with high transmittance which is good. Actually in real application, high transparency is not necessary and hard to keep it for long time. Why authors think this is very important? Does this treatment work for other materials?
2. For the wireless transmission, I see wires from fingers to controller board PCB in Fig.4a and details in Fig.4c, self-power means sensing signal to circuit as trigger or the power can driven the circuits on PCB for wireless transmission? Without advanced power management strategy, it is impossible to power this wireless module. And How about the transmission distance?

Responses to Reviewers

We sincerely appreciate the reviewer's valuable comments on our manuscript. The followings are point-by point responses to the reviewer's comments and the changes in the revised manuscript. Sentences inserted into the manuscript are marked in red.

Reviewer #1

Comment: This manuscript reports a transparent and stretchable triboelectric nanogenerator as self-powered and attachable ionic communicators. The authors need to address the following concerns when submitting the revised version.

Response: We greatly appreciate the reviewer's efforts to review the manuscript. We have carefully considered your comments. Herein, we explain how we revised the paper based on those comments and recommendations.

Comment: From the supplementary figure 2, a wire is connected to the hydrogel inside of the PDMS, but there is no wire connection shown in figure 1 (a). Can the authors explain how exactly did the wire connected? Is it simply inserted into STAIC after process flow in figure 1 (a)?

Response: Thank you for your insightful comment. A Pt electrode was inserted through capsulated PDMS into hydrogels to measure electrical characteristic of a STAIC. We specified the wiring in the Methods section.

Modification/Addition

1. Addition to the Methods on page 17.

→ A Pt electrode was inserted through capsulated PDMS to the hydrogel to measure electrical

characteristic of STAICs.

Comment: In figure 2 (e), triboelectric outputs were measured after 1000 times of uniaxial stretching, which demonstrates the stability of the STAICs after different strain conditions. However, have the authors considered the triboelectric output under different strain? Since it is attached to human skin, this kind of situation should also be taken into consideration.

Response: Thank you for your meaningful comment. We investigated triboelectric output voltage under different strain of 100 %, 150 %, and 200 %, respectively in Supplementary Fig. 6. When 200 % of strain was applied, the output voltages slightly increased from 252 V to 277 V. The effect of applied strain can be interpreted as follows; electrostatic induction which is induced from a charged surface to an electrode can be increased when the distance between the surface and electrode becomes closer as reported by Parida et al.¹ and Lai et al.² Because a uniaxial strain reduces the thickness of the sample by the Poisson's effect, induced output voltage was increased.

Modification/Addition

1. Modification of the Manuscript on page 8.

The STAICs do not show any mechanical fatigue or degradation in electrical output after being strained 1000 times with strains of 100 %, 150 %, and 200 % (Fig. 2e).

- Electrical outputs of the STAICs were examined under various uniaxial strains as shown in Supplementary Fig. 6. When 200 % of strain was applied, the output voltages slightly increased from 252 V to 277 V because of the reduced distance between the charged surface and the electrode. Degradations in electrical outputs by mechanical fatigues were tested after being strained 1000 times with strains of 100 %, 150 %, and 200 %. The STAICs do not show any degradation in electrical output by mechanical fatigues as shown in Fig. 2e.

2. Addition to the Supplementary Information on page 6..

Supplementary Figure 6 | Triboelectric outputs measured under 100 %, 150 % and 200 % of applied tensile strain.

We investigated triboelectric output voltages under different strains of 100 %, 150 % and 200 %. The measured output voltages slightly increased from 252 V to 277 V when 200 % of strain was applied. The effect of applied strain can be interpreted as follows; electrostatic induction which is induced from a charged surface to an electrode can be increased when the distance between the surface and electrode becomes closer as reported by Parida et al.¹ and Lai et al.² Because a uniaxial strain reduces the thickness of the sample by the Poisson's effect, induced output voltage was increased.

Comment: For voltage measurement, the authors use an oscilloscope (Tektronix, MDO3052).

What is the impedance of the measurement setup, since the impedance of TENG is high.

Response: We are grateful for your insightful questions. As shown in Fig. 2h, the inherent impedance of the STAICs estimated by Thévenin's theorem is about 13 MΩ, but the oscilloscope (Tektronix, MDO3052) that we have used to measure the output voltages has impedance of 1 MΩ. To increase the total impedance of the measurement setup, we have used high impedance voltage probes (Tektronix, P5100A) which have impedance of 40 MΩ. Because total impedance of our measurement setup was still not high enough, real output voltages induced by the STAICs could be larger than the measured values in our manuscript. As Wang and colleagues have used, an electrometer which has extremely high impedance could help to measure the output voltage

accurately³. We included the details about the measurement setup into the Methods section.

Modification/Addition

1. Addition to the Methods on page 17.

The electrical voltage and current of STAICs were measured by an oscilloscope (Tektronix, MDO3052) and a low-noise current preamplifier (Stanford research systems, SR570), respectively.

- The electrical voltage and current of STAICs were measured by using an oscilloscope (**1 MΩ, MDO3052, Tektronix**) with a voltage probe (**40 MΩ, TPP0850, Tektronix**) and a low-noise current preamplifier (Stanford research systems, SR570), respectively.

Comment: Supplementary Figure 3 shows that STAICs have good anti-dehydration ability in harsh conditions relative to the bare hydrogel, however we can still observe a 2 % reduction of the weight ratio after 5 hours. Does that mean that PDMS is not enough to perfectly seal the hydrogel to prevent the water evaporation? If so, did the authors tested the lifetime of the STAICs under harsh environment?

Response: Thank you for your consideration. Water permeability of PDMS is 2.63 ± 0.13 ($10^{-16} \text{ m}^2 \text{ s}^{-1} \text{ Pa}^{-1}$), which is higher than the value 0.4 of VHB (3M) and the value 0.026 of butyl rubber⁴. Water permeability of PDMS is not low enough to prevent evaporation of hydrogel perfectly. The result of evaporation tests which were performed under a vacuum of 0.013 atm at room temperature for a week is updated in Supplementary Fig. 5. Despite of high water permeability, STAICs didn't lose their triboelectric ability much even after a storage for one week in the harsh condition. Output voltages of STAICs were decreased about 10.9 % after being placed in a desiccator for a week.

Modification/Addition

1. Modification of Supplementary Figure 5.

Supplementary Figure 5 | Anti-dehydration ability of a STAIC under a vacuum of 0.013 atm at room temperature. **(a)** Normalized weight ratio of a STAIC and a hydrogel according to duration time. **(b)** Triboelectric output voltages of STAICs before and after being placed under the vacuum for a week. **(c)** Photographs showing contrasted extent of shrinkages of hydrogel and STAIC during the evaporation experiments. (scale bar : 1 cm)

Comment: It is better to label the static contact angle and receding contact angle in Fig. 3, for some readers may be not familiar with it.

Response: Thank you for your suggestion. By applying the reviewer’s suggestion, we have included words of “Advancing” and “Receding” under the title of “Dynamic contact angle” in Fig. 3.

Modification/Addition

1. Modification of Figure 3.

Comment: The demonstration of wireless real-time communication is interesting, but the need for transparency in this application is not strong enough.

Response: Thank you for your comments. High transparency of STAICs allows to generate electrical power without impeding optical information. Therefore, STAICs can be integrated with applications such as display panels, touch screen devices and so on. As shown in Supplementary Fig. 7, we have demonstrated the need for transparency by attaching a STAIC on top of a touch screen (this time, a resistive touch screen of an oscilloscope was used). A STAIC generated electrical power when the screen was touched, but they did not interrupt original functions of the touch screen because the STAIC are highly transparent. Furthermore, electrical power additionally generated by the STAIC could be more beneficial if the device has limited energy supplies like a portable device.

Modification/Addition

1. Addition to the Manuscript on page 9.

High transparency of STAICs allows to generate electrical power without impeding optical

information. We have demonstrated the need for transparency by attaching a STAIC on top of a touch screen (Supplementary Fig. 7). A STAIC generated electrical power when the screen was touched but did not hide the screen because the STAIC is highly transparent.

2. Addition to the Supplementary Information on page 7.

Supplementary Figure 7 | Electrical performance when touching (a) outside and (b) inside of STAIC boundary which is attached on a touch screen.

A STAIC was attached on top of a touch screen (this time, a resistive touch screen of an oscilloscope was used). A STAIC generated a voltage of 30 V when the screen was touched but did not hide the screen because the STAIC is highly transparent. Furthermore, electrical power additionally generated by the STAICs could be more beneficial if the device has limited energy supplies like a portable device.

Comment: The supplementary Fig 2 does not give any novel information. It is well know phenomenon.

Response: We appreciate your comment. As reviewer #1 mentioned, Supplementary Figure 2 does not contain new physics. However, because Nature Communications has a wide band of readers from various disciplines, explaining basic working mechanisms may help readers to understand the device. Furthermore, because reviewer #2 asked a few questions about Supplementary Fig. 2, we are

considering keeping the basic working mechanism in Supplementary Information. We would like to ask an opinion of reviewer #1 about this point again.

Comment: Please go through the manuscript more carefully since there are some grammar errors and typos, e.g. Page 10, Line 191: “this HDFs surface treatment has an advantage in its the transparency”; Page 10, Line 197: “By using ImageJ, photographs with dimensions of 30 by 30 mm” better be “30 mm by 30 mm”; sometimes “Fig. 3h iii” with “.” is used, sometimes “Fig 3h iv” is used, please be consistent; etc.

Response: Thank you very much for the corrections. We modified the grammatical errors and the typos as follows.

Modification/Addition

1. Modification of the Manuscript on page 10.

Compared to previous studies on modified surfaces with nano-micro patterning⁶, this HDFs surface treatment has an advantage in its the transparency.

- ➔ Compared to previous studies on modified surfaces with nano-micro patterning⁶, this HDFs surface treatment has an advantage in **keeping its transparency**.

2. Modification of the Manuscript on page 10.

By using ImageJ, photographs with dimensions of 30 by 30 mm

- ➔ By using ImageJ, photographs with dimensions of 30 **mm** by 30 mm

3. Modification of the Manuscript on page 11.

(Fig 3h iv)) and 5 minutes (Fig 3h v)).

- ➔ (Fig. 3h iv)) and 5 minutes (Fig. 3h v))

4. Modification of the Supplementary Information on page 5.

In the desiccator

→ In **a** desiccator

5. Modification of the Manuscript on page 6.

Method

→ **Methods**s

6. Modification of the Manuscript on page 6.

Experimental Section

→ **Methods**

7. Modification of the Manuscript on page 13.

For writing “R”, 18 is needed

→ For writing “R”, **a value of** 18 is needed

Comment: Figure 4 (c) shows the block diagram of self-powered communicator, in which the STAIC is connected to a controller board. Is the controller board powered by external source? If so, it might not be accurate to claim the whole system is self-powered. Please clearly illustrate this part in the manuscript. Besides, please provide the detail circuit diagram of the controller board in supplementary.

Response: Thank you very much for the valid comments. As the reviewer #1 pointed out, microcontroller and Wi-Fi module need an external power source. Although our research scope is limited to transmitting signals to a controller board by self-power generation without need for continuous power supply, because the entire system of a communicator is not fully self-powered, we have removed the word of “Self-powered” from the title to avoid confusions. Accordingly, the manuscript was also modified.

We have included a detailed circuit diagram of the controller board in Supplementary Fig. 10. Additional explanations about the flowchart of communication processing were included in Supplementary Fig. 11, too.

Modification/Addition

1. Modification of the title on page 1.

Self-powered, transparent, and attachable ionic communicators based on self-cleanable triboelectric nanogenerators.

- **Self-cleanable**, transparent and attachable ionic communicators based on triboelectric nanogenerators.

2. Modification of the Manuscript on page 2.

self-powered, transparent, and attachable ionic communicators (STAICs) based on self-cleanable triboelectric nanogenerators

- **self-cleanable**, transparent and attachable ionic communicators (STAICs) based on triboelectric nanogenerators

3. Modification of the Manuscript on page 5.

self-powered, transparent, and attachable ionic communicators

- **self-cleanable**, transparent and attachable ionic communicators

4. Modification of the Manuscript on page 8.

Self-powering mechanism and characterization

→ **Triboelectric** mechanism and characterization

5. Modification of the Manuscript on page 12.

we checked the self-powering property of STAICs on a finger by touching skin and clothes.

→ we checked the **triboelectric** property of STAICs on a finger by touching skin and clothes.

6. Modification of the Manuscript on page 12.

We then fabricated a circuit that can make real-time communication between the STAICs and a wireless computer as shown in Fig. 4c.

→ We then fabricated a circuit that can make real-time communication between the STAICs and a wireless computer as shown in Fig. 4c **and Supplementary Fig. 10.**

7. Modification of the Methods on page 17.

Through the low pass RC filter, it was connected to a channel of the microcontroller (Arduino uno, Arduino Co) with a Wi-Fi shield (SOLLAE SYSTEMS Co, PHPoC).

→ Through the low pass RC filter, it was connected to a channel of the microcontroller (Arduino uno, Arduino Co) with a Wi-Fi shield (SOLLAE SYSTEMS Co, PHPoC). **The microcontroller and the Wi-Fi shield were connected to a battery.**

8. Modification of the Methods on page 17.

Wireless self-powered real-time communication

→ Wireless real-time communication

9. Modification of the caption on page 23.

Self-powered, transparent, and attachable ionic communicators with high mechanical reliability.

→ **Self-cleanable**, transparent and attachable ionic communicators with high mechanical reliability

10. Modification of the caption on page 24.

Self-powering characterization

→ **Triboelectric** characterization

11. Modification of the caption on page 27.

Block diagram of self-powered communicator based on STAICs.

→ Block diagram of **the** communicator based on STAICs.

12. Addition to the Supplementary Information on page 10.

Supplementary Figure 10 | A detailed circuit diagram of the self-powered sensors and controller board.

13. Modification of the Supplementary Information on page 11.

To communicate between the combination of finger touches and the computer, coding was conducted. First, the micro-controller collects the input signal from each STAIC. Then micro-controller checks whether the signals are above the threshold value. Additionally, the micro-controller counts the number of consecutive samples that are above the threshold value in order to minimize the electrical signal noise. If the number of consecutive samples are above the required number of counts, the micro-controller sets the bit and collect every bit in the binary number corresponding to each STAIC and converts the binary number into the decimal number. Then, the micro-controller converts the binary number into the corresponding decimal number and maps the decimals to the corresponding letter.

→ A simple algorithm was developed and realized through C programming to convert a combination of finger touches into a letter and, further, words with meaning. First, the micro-controller collects the input signal, **voltage**, from each STAIC. **The** micro-controller checks whether the signals are above the threshold **voltage** value. Additionally, the micro-controller counts the number of consecutive samples that are above the threshold value in order to **filter out** the electrical signal noise. If the number of consecutive samples **from an input port is** above the required number of counts, the micro-controller **judges that the signals are valid and** sets the bit **corresponding to the input port.** If the number of consecutive samples is below the boundary, the signals are determined as electrical noise, and the bit is not set. A combination of these binary bits corresponding to input ports can generate thirty-one different cases which are thirty-one different letters in our case. **The micro-controller continually checks each** bit corresponding to each STAIC and **uses these bits to** converts the binary number into the decimal number. **Finally,** the micro-controller maps **the converted** decimal number to the corresponding letter.

Comment: Sweat is always a critical issue that affects the performance of the human-machine interfaces. In that case, did the author investigate the effects of humidity on the triboelectric output? This phenomenon may lead to a high instability of the device performance.

Response: Thank you for your valid comments. As advised by the reviewer #1, moisture has an impact on the triboelectric output^{5,6}. We also have checked the effects of humidity on the triboelectric output. The output voltage decreased from about 270 V to 50 V as the relative humidity increased from 20 % to 80 %. However, since it can still generate 50 V at a high relative humidity of 80 %, a STAIC could be operated under a high relative humidity of 80%.

Modification/Addition

1. Modification of the Manuscript on page 12.

In addition, the effect of humidity on triboelectric performance was investigated in Supplementary Fig. 9. The output voltage decreased from about 270 V to 50 V as the relative humidity increased from 20 % to 80 %. However, a STAIC can operate under a high relative humidity of 80%, because it can still generate detectable voltages.

2. Addition to the Supplementary Information on page 9.

Supplementary Figure 9 | (a) Measured output voltages and (b) averaged peak to peak voltages under humid conditions.

Reviewer #2

Comment: The author reported in the manuscript triboelectric nanogenerators. The device is transparent and can be easily cleaned. The property and performance have been investigated in detail. A self-powered communicator has also been fabricated and demonstrated. This work is important and of interests to others in the community. This reviewer supports the publication of this work, although a few questions are also listed below.

Response: We greatly appreciate the reviewer's efforts to review the manuscript. We have been considering a lot about practical application of a STAIC, and we are pleased that reviewer #2 gave a good evaluation of this part. We have carefully considered your comments. Herein, we explain how we revised the paper based on those comments and recommendations.

Comment: The author made a significant effort in studying the transparency of the device. It will be very important to explain why the transparency is important for the reported nanogenerator.

Response: Thank you for your comments. High transparency of STAICs allows to generate electrical power without impeding optical information. Therefore, STAICs can be integrated with applications such as display panels, touch screen devices and so on. As shown in Supplementary Fig. 7, we have demonstrated the need for transparency by attaching a STAIC on top of a touch screen (this time, a resistive touch screen of an oscilloscope was used). A STAIC generated electrical power when the screen was touched, but they did not interrupt original functions of the touch screen because the STAIC are highly transparent. Furthermore, electrical power additionally generated by the STAIC could be more beneficial if the device has limited energy supplies like a portable device.

Modification/Addition

1. Addition to the Manuscript on page 9.

High transparency of STAICs allows to generate electrical power without impeding optical information. We have demonstrated the need for transparency by attaching a STAIC on top of a touch screen (Supplementary Fig. 7). A STAIC generated electrical power when the screen was touched but did not hide the screen because the STAIC is highly transparent.

2. Addition to the Supplementary Information on page 7.

Supplementary Figure 7 | Electrical performance when touching (a) outside and (b) inside of STAIC boundary which is attached on a touch screen.

A STAIC was attached on top of a touch screen (this time, a resistive touch screen of an oscilloscope was used). A STAIC generated a voltage of 30 V when the screen was touched but did not hide the screen because the STAIC is highly transparent. Furthermore, electrical power additionally generated by the STAICs could be more beneficial if the device has limited energy supplies like a portable device.

Comment: The nanogenerator worked in a single electrode mode. The contact material would be the other electrode. It will be helpful to study or discuss the effect of contact materials.

Response: Thank you for the constructive suggestion. One of main working principles of triboelectric nanogenerators is contact electrification based on relative difference of electron affinity of materials, and it has been reported by previous researchers⁷. The greater difference of electron affinity between the materials makes the larger contact electrification, which leads to higher output

voltage. We have performed additional experiments, using aluminum, copper, polyethylene terephthalate (PET), polyimide (PI) and polytetrafluoroethylene (PTFE) films as contact materials. Supplementary Figure 3a shows relative electron affinity of materials we have used. Greater electron affinity of the STAIC than that of contact materials led larger output voltages (Supplementary Fig. 3b). In case of contact with PTFE, generated voltage had opposite direction from those of other materials. It can be attributed to greater electron affinity of PTFE than that of a STAIC.

Modification/Addition

1. Addition to the Manuscript on page 8.

We chose an aluminium plate as a contact material. Different contact materials were investigated in Supplementary Fig. 3. Larger difference between electron affinities of contact materials led to higher triboelectric voltage outputs.

2. Addition to the Methods on page 17.

A pushing tester (Junil Tech, JIPT-100) was used to create vertical compressive force

- A pushing tester (Junil Tech, JIPT-100) was used to create vertical compressive force with an aluminium plate as a contact material.

3. Addition to the Supplementary Information on page 3.

Supplementary Figure 3 | Triboelectric performances of a STAIC were investigated with various contact materials. **(a)** Triboelectric series. **(b)** Output voltages from a STAIC when touched with different materials.

We measured output voltages of a STAIC, using aluminum, copper, polyethylene terephthalate (PET), polyimide (PI) and polytetrafluoroethylene (PTFE) films as contact materials. Supplementary Figure 3a shows relative electron affinity of materials we used. Larger difference between electron affinities of contact materials led to higher triboelectric voltage outputs. When PTFE was used as a contact material, the sign of generated voltage was flipped. It can be attributed to greater electron affinity of PTFE than that of a STAIC.

Comment: Again, the performance should be affected by the contact material (fabric, skin, or others). These materials are different in keep static charges. How does the energy conversion process change when time is considered. In other words, if the supplementary figure 2c stay for a longer time, the charges on the finger might changes, such that the charge transport (energy conversion) in supplementary figure 2d might be different.

Response: Thank you for your careful advice. In order to check the effect of time interval between

consecutive contacts on energy conversion, we performed experiments in which we varied duration time from 1 s to 512 s. The output voltage decreased as the duration time increased, and the output voltages with 512 sec duration become almost zero. The results may originate from charge leakages to the air.

Modification/Addition

1. Addition to the Manuscript on page 8.

In addition, effect of time interval between consecutive contacts on the electrical output performance was investigated (Supplementary Fig. 4). Longer duration time after detachment caused lower output voltages.

2. Addition to the Supplementary Information on page 4.

Supplementary Figure 4 | (a) The effect of time interval between consecutive contacts on the electrical output performances and (b) averaged voltages.

In order to check the effect of time interval between consecutive contacts on energy conversion, we performed experiments in which we varied duration time from 1 s to 512 s (Supplementary Fig. 4a). An increase in time interval between consecutive contacts caused output voltage to decrease, a result which comes from charge leakage as shown in Supplementary Fig. 4b.

Comment: The explanation of the self-powered communicator is not clear. What is the circuit? How the communication realized?

Response: Thank you very much for the valid comments. We have included a detailed circuit diagram of the controller board in Supplementary Fig. 10. Additional explanations about the flowchart of communication processing were included in Supplementary Fig. 11, too.

Modification/Addition

1. Modification of the Manuscript on page 12.

We then fabricated a circuit that can make real-time communication between the STAICs and a wireless computer as shown in Fig. 4c.

- We then fabricated a circuit that can make real-time communication between the STAICs and a wireless computer as shown in Fig. 4c and **Supplementary Fig. 10**.

2. Addition to the Supplementary Information on page 10.

Supplementary Figure 10 | A detailed circuit diagram of the self-powered sensors and controller board.

3. Modification of the Supplementary Information on page 11.

To communicate between the combination of finger touches and the computer, coding was conducted. First, the micro-controller collects the input signal from each STAIC. Then micro-controller checks whether the signals are above the threshold value. Additionally, the micro-controller counts the number of consecutive samples that are above the threshold value in order to minimize the electrical signal noise. If the number of consecutive samples are above the required number of counts, the micro-controller sets the bit and collect every bit in the binary number corresponding to each STAIC and converts the binary number into the decimal number. Then, the micro-controller converts the binary number into the corresponding decimal number and maps the decimals to the corresponding letter.

- A simple algorithm was developed and realized through C programming to convert a combination of finger touches into a letter and, further, words with meaning. First, the micro-controller collects the input signal, **voltage**, from each STAIC. **The** micro-controller checks whether the signals are above the threshold **voltage** value. Additionally, the micro-controller counts the number of consecutive samples that are above the threshold value in order to **filter out** the electrical signal noise. If the number of consecutive samples **from an input port is** above the required number of counts, the micro-controller **judges that the signals are valid and** sets the bit **corresponding to the input port.** If the number of consecutive samples is below the boundary, the signals are determined as electrical noise, and the bit is not set. A combination of these binary bits corresponding to input ports can generate thirty-one different cases which are thirty-one different letters in our case. **The micro-controller continually checks each** bit corresponding to each STAIC and **uses these bits to** converts the binary number into the decimal number. **Finally,** the micro-controller maps **the converted** decimal number to the corresponding letter.

Reviewer #3

Comment: This manuscript presents a novel transparent attachable ionic communicator driven by TENGs. The originality of this paper is significant, application of self-cleaning effect is smart. The result is interesting and promising for five fingers communicator. However, some results are not well explained in the text. The following aspects should be revised before considering this manuscript for publication.

Response: Thank you very much for the reviewer's efforts to review the manuscript. We have made efforts to create a practical ionic communicator, and we are pleased that reviewer #3 gave a good evaluation of this part. We have carefully considered your comments. Herein, we explain how we revised the paper based on those comments and recommendations.

Comment: In the manuscript, the authors illustrate that hydrogel by HDFs treatment to provide high electrification with high transmittance which is good. Actually in real application, high transparency is not necessary and hard to keep it for long time. Why authors think this is very important? Does this treatment work for other materials?

Response: Thank you for your comments. High transparency of STAICs allows to generate electrical power without impeding optical information. Therefore, STAICs can be integrated with applications such as display panels, touch screen devices and so on. As shown in Supplementary Fig. 7, we have demonstrated the need for transparency by attaching a STAIC on top of a touch screen (this time, a resistive touch screen of an oscilloscope was used). A STAIC generated electrical power when the screen was touched, but they did not interrupt original functions of the touch screen because the STAIC are highly transparent. Furthermore, electrical power additionally generated by the STAIC could be more beneficial if the device has limited energy supplies like a portable device.

Materials with hydroxyl groups ($-OH$) can be coated with HDFs. In this research, because of plasma treatment, hydroxyl groups are formed on PDMS surface. Hydroxyl groups on the PDMS

surface form strong covalent bonds with the chlorosilyl groups of the HDFs. As a result, chemically assembled monolayer with a fluorocarbon chain could be formed on the surface. Trifluoromethyl groups ($-\text{CF}_3$) are arranged along the outer surface⁸.

Modification/Addition

1. Addition to the Manuscript on page 9.

High transparency of STAICs allows to generate electrical power without impeding optical information. We have demonstrated the need for transparency by attaching a STAIC on top of a touch screen (Supplementary Fig. 7). A STAIC generated electrical power when the screen was touched but did not hide the screen because the STAIC is highly transparent.

2. Addition to the Supplementary Information on page 7.

Supplementary Figure 7 | Electrical performance when touching (a) outside and (b) inside of STAIC boundary which is attached on a touch screen.

A STAIC was attached on top of a touch screen (this time, a resistive touch screen of an oscilloscope was used). A STAIC generated a voltage of 30 V when the screen was touched but did not hide the screen because the STAIC is highly transparent. Furthermore, electrical power additionally generated by the STAICs could be more beneficial if the device has limited energy supplies like a portable device.

Comment: For the wireless transmission, I see wires from fingers to controller board PCB in Fig.4a and details in Fig.4c, self-power means sensing signal to circuit as trigger or the power can driven the circuits on PCB for wireless transmission? Without advanced power management strategy, it is impossible to power this wireless module. And How about the transmission distance?

Response: Thank you very much for the valid comments. As the reviewer #3 pointed out, microcontroller and Wi-Fi module need an external power source. Although our research scope is limited to transmitting signals to a controller board by self-power generation without need for continuous power supply, because the entire system of a communicator is not fully self-powered, we have removed the word of “Self-powered” from the title to avoid confusions. Accordingly, the manuscript was also modified.

We have included a detailed circuit diagram of the controller board in Supplementary Fig. 10. Additional explanations about the flowchart of communication processing were included in Supplementary Fig. 11, too.

Transmission distance depends on Wi-Fi hardware. Specification sheets of commercialized Wi-Fi adapters do not include the information of transmission distance since it is affected by various factors, such as spatial obstacles (eg. walls), weather, and the number of users. To check the maximum communication distance of STAIC, we tested the Wi-Fi adapter (GWF-3S03) and verified that Wi-Fi was stably connected within a range of 10 meter distance when the module was not blocked by any objects.

Modification/Addition

1. Modification of the title on page 1.

Self-powered, transparent, and attachable ionic communicators based on self-cleanable triboelectric nanogenerators.

- **Self-cleanable**, transparent and attachable ionic communicators based on triboelectric nanogenerators.

2. Modification of the Manuscript on page 2.

self-powered, transparent, and attachable ionic communicators (STAICs) based on self-cleanable triboelectric nanogenerators

→ **self-cleanable**, transparent and attachable ionic communicators (STAICs) based on triboelectric nanogenerators

3. Modification of the Manuscript on page 5.

self-powered, transparent, and attachable ionic communicators

→ **self-cleanable**, transparent and attachable ionic communicators

4. Modification of the Manuscript on page 8.

Self-powering mechanism and characterization

→ **Triboelectric** mechanism and characterization

5. Modification of the Manuscript on page 12.

we checked the self-powering property of STAICs on a finger by touching skin and clothes.

→ we checked the **triboelectric** property of STAICs on a finger by touching skin and clothes.

6. Modification of the Manuscript on page 12.

We then fabricated a circuit that can make real-time communication between the STAICs and a wireless computer as shown in Fig. 4c.

→ We then fabricated a circuit that can make real-time communication between the STAICs and a wireless computer as shown in Fig. 4c **and Supplementary Fig. 10.**

7. Modification of the Methods on page 17.

Through the low pass RC filter, it was connected to a channel of the microcontroller (Arduino uno, Arduino Co) with a Wi-Fi shield (SOLLAE SYSTEMS Co, PHPoC).

- Through the low pass RC filter, it was connected to a channel of the microcontroller (Arduino uno, Arduino Co) with a Wi-Fi shield (SOLLAE SYSTEMS Co, PHPoC). **The microcontroller and the Wi-Fi shield were connected to a battery.**

8. Modification of the Methods on page 17.

Wireless self-powered real-time communication

- Wireless real-time communication

9. Modification of the caption on page 23.

Self-powered, transparent, and attachable ionic communicators with high mechanical reliability.

- **Self-cleanable**, transparent and attachable ionic communicators with high mechanical reliability

10. Modification of the caption on page 24.

Self-powering characterization

- **Triboelectric** characterization

11. Modification of the caption on page 27.

Block diagram of self-powered communicator based on STAICs.

- Block diagram of **the** communicator based on STAICs.

12. Addition to the Supplementary Information on page 10.

Supplementary Figure 10 | A detailed circuit diagram of the self-powered sensors and controller board.

13. Modification of the Supplementary Information on page 11.

To communicate between the combination of finger touches and the computer, coding was conducted. First, the micro-controller collects the input signal from each STAIC. Then micro-controller checks whether the signals are above the threshold value. Additionally, the micro-controller counts the number of consecutive samples that are above the threshold value in order to minimize the electrical signal noise. If the number of consecutive samples are above the required number of counts, the micro-controller sets the bit and collect every bit in the binary number corresponding to each STAIC and converts the binary number into the decimal number. Then, the micro-controller converts the binary number into the corresponding decimal number and maps the decimals to the corresponding letter.

→ A simple algorithm was developed and realized through C programming to convert a combination of finger touches into a letter and, further, words with meaning. First, the micro-controller collects the input signal, **voltage**, from each STAIC. **The** micro-controller checks whether the signals are above the threshold **voltage** value. Additionally, the micro-

controller counts the number of consecutive samples that are above the threshold value in order to **filter out** the electrical signal noise. If the number of consecutive samples **from an input port is** above the required number of counts, the micro-controller **judges that the signals are valid and** sets the bit **corresponding to the input port**. If the number of consecutive samples is below the boundary, the signals are determined as electrical noise, and the bit is not set. A combination of these binary bits corresponding to input ports can generate thirty-one different cases which are thirty-one different letters in our case. The micro-controller **continually checks each** bit corresponding to each STAIC and **uses these bits to** converts the binary number into the decimal number. **Finally**, the micro-controller maps **the converted** decimal number to the corresponding letter.

1. Parida K, Kumar V, Jiangxin W, Bhavanasi V, Bendi R, Lee PS. Highly Transparent, Stretchable, and Self-Healing Ionic-Skin Triboelectric Nanogenerators for Energy Harvesting and Touch Applications. *Advanced Materials* **29**, (2017).
2. Lai YC, *et al.* Electric Eel-Skin-Inspired Mechanically Durable and Super-Stretchable Nanogenerator for Deformable Power Source and Fully Autonomous Conformable Electronic-Skin Applications. *Advanced Materials* **28**, 10024-10032 (2016).
3. Wang X, *et al.* Self-powered high-resolution and pressure-sensitive triboelectric sensor matrix for real-time tactile mapping. *Advanced materials* **28**, 2896-2903 (2016).
4. Le Floch P, *et al.* Wearable and washable conductors for active textiles. *ACS Applied Materials & Interfaces* **9**, 25542-25552 (2017).
5. Lee KY, *et al.* Hydrophobic Sponge Structure-Based Triboelectric Nanogenerator. *Advanced materials* **26**, 5037-5042 (2014).
6. Guo H, Chen J, Tian L, Leng Q, Xi Y, Hu C. Airflow-induced triboelectric nanogenerator as a self-powered sensor for detecting humidity and airflow rate. *ACS applied materials & interfaces* **6**, 17184-17189 (2014).
7. Wang ZL. Triboelectric nanogenerators as new energy technology for self-powered systems and as active mechanical and chemical sensors. *ACS nano* **7**, 9533-9557 (2013).
8. Lee S, Kim W, Lee S, Shim S, Choi D. Controlled transparency and wettability of large-area nanoporous anodized alumina on glass. *Scripta Materialia* **104**, 29-32 (2015).

Reviewers' Comments:

Reviewer #2 (Remarks to the Author):

The author addressed most concerns of this reviewer. The author might want to consider the following suggestions before publishing this work.

1. "Self-cleanable" was added to the manuscript. It is not clear why "self-cleanable" is important for this work, and the manuscript does not provide enough evidence to show the device is "self-cleanable".
2. The manuscript includes many supplementary figures. It will be better to include briefly the major result of these supplementary works. For example, on page 8, the author wrote "effect of time interval between consecutive contacts on the electrical output performance was investigated (Supplementary Fig. 4)." It will be much more helpful and informative if the author simply provides the major conclusion from this investigation.

Reviewer #3 (Remarks to the Author):

Authors addressed my concerns clearly, the modification is high quality, can be published as it is.

Responses to Reviewers

We sincerely appreciate the reviewer's valuable comments on our manuscript. The following are point-by-point responses to the reviewer's comments and the changes in the revised manuscript.

Reviewer #2

Comment: The author addressed most concerns of this reviewer. The author might want to consider the following suggestions before publishing this work.

Response: Thank you very much for the reviewer's efforts to review the manuscript. We have made efforts to answer the questions reviewer #2 mentioned, and we are pleased that reviewer #2 gave a good evaluation. We have carefully considered your comments. Herein, we explain how we revised the paper based on those comments and recommendations.

Comment: "Self-cleanable" was added to the manuscript. It is not clear why "self-cleanable" is important for this work, and the manuscript does not provide enough evidence to show the device is "self-cleanable".

Response: Thank you very much for the reviewer's considerable comment. Self-cleanability is defined as ability of contamination-free to remove any debris from their surfaces in a variety of ways (doi: 10.1146/annurev-matsci-070511-155046). Contact angle hysteresis (C.A.H) is a major influence on self-cleanability. C.A.H difference depending on the surface treatment of STAICs has been investigated. Figure 3d, e, f and Supplementary Fig. 8 show that surface treatment makes an effect on

reducing contact angle hysteresis and increases self-cleanability.

TENGs with polymer surface might easily get contaminated during operations resulting in a decrease of electrical output performances and transparency. Self-cleanability is important for electrical and optical stable TENGs to avoid the problems as shown in Figure 3k-m.

Comment: The manuscript includes many supplementary figures. It will be better to include briefly the major result of these supplementary works. For example, on page 8, the author wrote "effect of time interval between consecutive contacts on the electrical output performance was investigated (Supplementary Fig. 4)." It will be much more helpful and informative if the author simply provides the major conclusion from this investigation.

Response: We appreciate your kind comment. We added brief conclusion to the manuscript.

Modification/Addition

1. Addition to the manuscript on page 8.

, a result which comes from charge leakage.

2. Addition to the manuscript on page 8.

Bare hydrogel noticeably shrunk after 3 hours in a desiccator and its weight decreased by 72.5 %, while STAIC decreased by only 2.8 %.

Reviewer #3

Comment: Authors addressed my concerns clearly, the modification is high quality, can be published as it is.

Response: We greatly appreciate the reviewer's efforts to review the manuscript. We have been considering a lot about the reviewer's comments and recommendations, and we are pleased that reviewer #3 gave a good evaluation of this work.